# *Chrysanthemi Zawadskii* var. *Latilobum* Attenuates Obesity-Induced Skeletal Muscle Atrophy via Regulation of PRMTs in Skeletal Muscle of Mice

**DOI:** 10.3390/ijms21082811

**Published:** 2020-04-17

**Authors:** Ahyoung Yoo, Young Jin Jang, Jiyun Ahn, Chang Hwa Jung, Hyo Deok Seo, Tae Youl Ha

**Affiliations:** 1Division of Food Functionality Research, Korea Food Research Institute, Wanju-gun 55365, Korea; 50026@kfri.re.kr (A.Y.); jyj616@kfri.re.kr (Y.J.J.); jyan@kfri.re.kr (J.A.); chjung@kfri.re.kr (C.H.J.); hyo-deok.seo@kfri.re.kr (H.D.S.); 2Department of Food and Nutrition, Chungnam National University, Daejeon 34134, Korea; 3Division of Food Biotechnology, University of Science and Technology, Daejeon 34113, Korea

**Keywords:** *Chrysanthemi zawadskii* var. *latilobum*, obesity-induced skeletal muscle atrophy, PRMTs, mitochondrial function

## Abstract

As obesity promotes ectopic fat accumulation in skeletal muscle, resulting in impaired skeletal muscle and mitochondria function, it is associated with skeletal muscle loss and dysfunction. This study investigated whether *Chrysanthemi zawadskii* var. *latilobum* (CZH) protected mice against obesity-induced skeletal muscle atrophy and the underlying molecular mechanisms. High-fat diet (HFD)-induced obese mice were orally administered either distilled water, low-dose CZH (125 mg/kg), or high-dose CZH (250 mg/kg) for 8 w. CZH reduced obesity-induced increases in inflammatory cytokines levels and skeletal muscle atrophy, which is induced by expression of atrophic genes such as muscle RING-finger protein 1 and muscle atrophy F-box. CZH also improved muscle function according to treadmill running results and increased the muscle fiber size in skeletal muscle. Furthermore, CZH upregulated mRNA and protein levels of protein arginine methyltransferases (PRMT)1 and PRMT7, which subsequently attenuated mitochondrial dysfunction in the skeletal muscle of obese mice. We also observed that CZH significantly decreased PRMT6 mRNA and protein expression, which resulted in decreased muscle atrophy. These results suggest that CZH ameliorated obesity-induced skeletal muscle atrophy in mice via regulation of PRMTs in skeletal muscle.

## 1. Introduction

Skeletal muscle atrophy is a debilitating condition associated with decreased protein synthesis and the accelerated degradation of muscle fibers [1]. Reduced protein synthesis leads to sharp decreases in skeletal muscle mass, which may result in falls and possible bone fractures [2]. Degradation of muscle fibers is followed by reduced muscle strength and loss of mitochondria in myocytes [3]. Recently, skeletal muscle loss and dysfunction (physical strength, mobility, and vitality) have been associated with obesity, referred to as sarcopenic obesity [4]. Obesity promotes inflammation in adipose tissue, increased lipolysis, and subsequent ectopic fat accumulation in skeletal muscles, which results in impaired skeletal muscles [5]. In addition, the skeletal muscle of obese humans exhibits decreased mitochondrial content and reduced function of fatty acid oxidation [6].

Several transcriptional regulators have been reported to regulate muscle atrophy. For instance, peroxisome proliferator activated receptor gamma coactivator 1α (PGC1α) is the prime regulator of mitochondrial content and oxidative metabolism and is important for maintaining muscle energy homeostasis. PGC1α is also an important factor that opposes the effect of forkhead transcription factor3 (FOXO3), which regulates the transcription of muscle atrophy-related genes [7]. Interestingly, recent studies have shown that protein arginine methyltransferases (PRMTs) regulate PGC1α and FOXO3. PRMT1 coactivates nuclear receptors and has been reported to induce PGC1α function through the methylation of several arginine residues in the C-terminal region [8]. Meanwhile, PRMT7 regulates PGC1α expression via the activation of p38 mitogen-activated protein kinase (p38MAPK), likely through arginine methylation [9]. On the other hand, PRMT6 activates FOXO3 and the ubiquitin-proteasome degradation pathway in skeletal muscle [10].

*Chrysanthemi zawadskii* var. *latilobum* (CZH) is a perennial herb from the family Asteraceae. CZH is widely distributed in Asia and Northeastern Europe and has been used as a tea and as a traditional medicine in Korea and China as a treatment for inflammatory diseases, gastroenteric troubles, and uterine diseases such as menstrual irregularity and infertility [11,12]. The pharmacological attributes of CZH include anti-inflammatory, anti-oxidant [13,14,15], and hepatoprotective [16] effects. Interestingly, anti-hyperglycemia [17] and anti-obesity [18] effects of CZH have also been reported. Based on these studies, we hypothesized that CZH-induced changes in glucose and fatty acid metabolisms would attenuate muscle atrophy. Therefore, the current study was designed to investigate whether CZH affected obesity-induced skeletal muscle atrophy and to clarify the possible in vivo mechanism of CZH regulating PRMTs.

## 2. Results

### 2.1. CZH Ameliorated Obesity-Induced Skeletal Muscle Accumulation of Fat and Metabolic Parameters in Obese Mice

To investigate the effect of CZH on obesity-induced skeletal muscle atrophy, we measured muscle weight and skeletal muscle triglyceride (TG) content. Muscle weight was calculated as the sum weights of the tibialis anterior, gastrocnemius, quadriceps, and triceps muscles. High-fat diet (HFD)-fed mice had increased body weight and decreased muscle weight compared to those of the chow-fed control mice (Chow). The final body weight of CZH groups did not differ from that of the HFD group (Figure 1A). The CZH groups tended to have slightly increased muscle weight (Figure 1B) compared to the HFD group. Furthermore, obesity promoted ectopic fat accumulation in skeletal muscle. The HFD induced TG accumulation in skeletal muscle, but either 125 (125CZH) or 250 mg/kg CZH (250CZH) significantly reduced the HFD-induced TG accumulation in skeletal muscle (Figure 1C). As the skeletal muscle is the major site of insulin-stimulated glucose disposal [19], we measured serum glucose and insulin levels. The HFD-fed mice had more serum glucose and insulin levels than the chow-fed mice. However, CZH treatment reduced the HFD-induced increased levels of glucose and insulin (Figure 1D). Obesity results in increased circulating inflammatory mediators, leading to metabolic derangements within skeletal muscle [20]. The HFD-fed mice had increased levels in the serum and mRNA expression in skeletal muscle of inflammatory cytokines such as tumor necrosis factor-alpha (TNFα), monocyte chemoattractant protein 1 (MCP1), and interleukin 1 beta (IL-1β). CZH reduced the obesity-induced increased levels of the inflammatory cytokines in both the serum (Figure 1E) and skeletal muscle (Figure 1F). To examine the toxicity of CZH, we measured serum alanine aminotransferase (ALT) and aspartate aminotransferase (AST) levels. The serum levels of these enzymes were significantly increased in HFD-fed mice relative to the levels in the control group. CZH treatment attenuated the HFD-induced increase in serum ALT and AST levels (Figure 1G). These results indicate that CZH ameliorated the obesity-induced fat accumulation and metabolic parameters in obese mice.

### 2.2. CZH Inhibited Obesity-Induced Skeletal Muscle Atrophy in HFD-Fed Mice

Two muscle-specific E3 ubiquitin ligases, muscle RING-finger protein 1 (MuRF1) and muscle atrophy F-box (Atrogin), are key regulators of ubiquitin-mediated protein degradation in skeletal muscle. We evaluated the effect of CZH on MuRF1 and Atrogin gene expression in skeletal muscle. HFD increased MuRF1 and Atrogin mRNA and protein levels compared to those of the Chow mice. However, CZH significantly decreased MuRF1 and Atrogin mRNA expression (Figure 2A), which correlated with decreases in MuRF1 and Atrogin protein levels (Figure 2B). The combination of decreased physical activity (running) and increased weakness (grip strength) of the obese mice fulfilled the criteria for skeletal muscle atrophy. We then evaluated the effect of CZH on exercise performance using treadmill running and grip strength. HFD decreased the running distance, time to exhaustion, and grip strength compared to those of the Chow mice. However, CZH increased the running distance and time to exhaustion (Figure 2C) and there was a trend toward increased grip strength in the CZH group (Figure 2D). These results indicate that CZH greatly improved the HFD-induced skeletal muscle atrophy in obese mice.

### 2.3. CZH Increased Muscle Fiber Size and Myosin Heavy Chain (MHC) Isoform in Obese Mice

Muscle atrophy results in a decreased cross-sectional area (CSA) of skeletal muscle. According to muscle CSA measurements, feeding mice an HFD decreased muscle fiber size compared to that of the Chow mice. However, CZH increased the CSA of skeletal muscle compared to that of the HFD group (Figure 3A). To determine whether CZH altered muscle fiber type, mRNA and protein levels of MHC in skeletal muscle were quantified by qRT-PCR and Western blotting. As shown in Figure 3B,C, mRNA expressions of MHC1 and MHC2A, respectively, in the 250CZH group were significantly increased compared to those of the HFD group. However, mRNA expression levels of MHC2B tended to increase (Figure 3B). Total MHC, MHC1, MHC2A, and MHC2B protein levels were significantly increased in skeletal muscle of the 250CZH group compared to those of the HFD group (Figure 3C). These results indicate that CZH increased the muscle fiber size and the amount of MHC muscle fiber isoform in HFD-fed mice.

### 2.4. CZH Regulated Skeletal Muscle PRMTs in Obese Mice

PRMT1 and PRMT7 have been reported to play critical roles in PGC1α function [8,9], which increases mitochondrial biogenesis and fatty acid β-oxidation through methylation of several arginine residues. We observed that CZH supplementation increased PRMT1, PRMT7, and PGC1α mRNA expression (Figure 4A), which translated into increased PRMT1, PRMT7, and PGC1α protein levels (Figure 4B). We also examined the expression levels of nuclear respiratory factor 1 (NRF1), nuclear factor erythroid 2-related factor 2 (NRF2), kelch-like ECH-associated protein 1 (Keap1), mitochondrial transcription factor A (TFAM), estrogen-related receptor gamma (ERRγ), and PPARδ in skeletal muscle, which are known to be regulated by PGC1α. Expression levels of NRF1, NRF2, and TFAM mRNAs were significantly elevated by CZH supplementation, and the expression of ERRγ, and PPARδ mRNA tended to increase. However, the mRNA level of Keap1, a negative regulator of NRF2, was significantly reduced in the CZH-treated group relative to the levels in the HFD group (Figure 4C). CZH supplementation also enhanced the protein expression of NRF1, NRF2, ERRγ, and PPARδ compared to those in HFD-fed mice (Figure 4D). PRMT6 activates FOXO3 and the ubiquitin-proteasome degradation pathway in skeletal muscle, resulting in muscle atrophy [10]. We observed that CZH significantly decreased PRMT6 mRNA levels and protein expression elevated by HFD (Figure 4E,F, respectively). Furthermore, levels of phospho-FOXO3 in skeletal muscle was reduced by CZH (Figure 4F). These results indicate that CZH regulated the skeletal muscle PRMTs in HFD-fed mice.

### 2.5. CZH Improved Mitochondrial Dysfunction in HFD-Fed Mice

Several studies have shown that skeletal muscle from obese mice exhibits reduced citrate synthase activity and those of several complexes of the electron transport chain, as well as decreased oxygen consumption and ATP production [21,22,23]. To investigate the effect of CZH on mitochondrial function, we measured the mitochondrial DNA (mtDNA) content and the activities of citrate synthases and mitochondrial respiratory complexes I and II in the skeletal muscle. As shown in Figure 5A, the CZH-treated mice tended to have more mtDNA content than HFD-fed mice. The citrate synthase and complex I and II subunit activities decreased significantly in the skeletal muscle of the HFD group. However, CZH improved citrate synthase and complex I and II subunit activities in the skeletal muscle (Figure 5B,C). We also analyzed mRNA levels of oxidative phosphorylation (OXPHOS) genes in skeletal muscle (Figure 5D). HFD-fed mice tended to have a decreased mRNA expression of OXPHOS genes in skeletal muscle. In contrast, the CZH groups tended to have an increased mRNA expression of OXPHOS genes in skeletal muscle compared to that of HFD-fed mice. Specifically, mRNA levels of the NADH:ubiquinone oxidoreductase subunit AB1 (NDUFAB1), cytochrome c (CYCS), ubiquinol-cytochrome C reductase core protein 1 (UQCRC1), and ubiquinol-cytochrome C reductase rieske iron-sulfur polypeptide 1 (UQCRFS1) complexes I and III in the muscle of the CZH group were significantly increased compared to those in the HFD-fed mice (Figure 5D). These results indicate that CZH increases mitochondrial oxidative capacity, leading to improvements in the mitochondrial dysfunction in HFD-fed mice.

### 2.6. CZH, Linarin, and Luteolin Attenuated Palmitic Acid-Induced Muscle Atrophy in C2C12 Cells

To investigate which component(s) of CZH contributed to the protective effect of CZH on obesity-induced muscle atrophy, we examined the effect of linarin and luteolin, which have been reported to be the bioactive components of CZH. Experiments were performed using palmitic acid (PA)-treated differentiated C2C12 myoblasts (myotubes) to model an environment similar to obesity. As shown in Figure 6, PA treatment of C2C12 cells increased the mRNA expression levels of MuRF1 and Atrogin genes compared to those of bovine serum albumin (BSA)-treated control cells. However, treatment with CZH, linarin, and luteolin significantly decreased MuRF1 and Atrogin mRNA expression in C2C12 cells.

## 3. Discussion

Sarcopenic obesity accompanied by combined abnormal muscle loss and the accumulation of body fat correlates with muscle strength and function and synergistically maximizes their health-threatening effects [24,25]. Recent studies suggest that diet-induced obesity alone may reprogram skeletal muscle to increase the production of inflammatory cytokines, including TNFα, MCP1, and IL-1β [26]. Elevated circulating TNFα in obese individuals can especially lead to muscle loss and inflammatory myopathies by regulating the activation and secretion of other inflammatory cytokines [27,28]. In the current study, HFD-induced obese mice exhibited increased levels of inflammatory cytokines levels and skeletal muscle atrophy, as indicated by the induction of atrophic genes MuRF1 and Atrogin and the reduction of skeletal muscle CSA. However, CZH groups demonstrated a reversal of the obesity-induced inflammation and skeletal muscle atrophy. Obesity is also known to further promote ectopic fat accumulation in skeletal muscle, but our findings showed that CZH significantly reduced TG accumulation in skeletal muscle.

Mitochondria are closely related to the function of the skeletal muscles, as these organelles constitute the main energy supply for contractile muscles [29]. For example, mitochondria are involved in cell death through the release of apoptogenic factors, which induce DNA condensation, DNA degradation, and apoptosome formation [30]. Elevated apoptosis in the skeletal muscle has been increasingly recognized to cause muscle atrophy [31]. Furthermore, mitochondria are a major source of cellular reactive oxygen species that emerge as superoxide molecules at various positions along the electron transport chain [32]. Oxidative stress has been suggested to be a key factor contributing to the initiation and progression of muscle atrophy [33].

The skeletal muscle of obese individuals characteristically shows lower mtDNA content and decreased citrate synthase activity [34,35]. Mitochondrial biogenesis is initiated with the increased transcription of both nuclear DNA and mtDNA [36]. PGC1α regulates mitochondrial biogenesis through regulation of NRF1 and NRF2 [37], which, in turn, regulate TFAM [38]. TFAM plays an important role in maintaining the mtDNA copy number and structure and is, therefore, crucial for efficient transcription of mtDNA, which encodes the core hydrophobic proteins involved in OXPHOS [39]. PGC-1α not only stimulates mitochondrial biogenesis through NRF1 and NRF2 expression, but also leads to activation of genes responsible for fatty acid oxidation through increased ERRγ and PPARδ expression. ERRγ, which is active even when PGC1α is not induced, shares many target genes with ERRα [40]. When PGC1 is induced, ERRα is the primary regulator of the mitochondrial biogenic gene network [41]. To induced fatty acid oxidation, PPARδ increases the proportion of oxidative fibers that are rich in mitochondria, thereby dramatically boosting mitochondrial oxidative metabolism. In the current study, HFD-induced obese mice exhibited decreased citrate synthase and mitochondrial respiratory complexes I and II activity in skeletal muscle. Furthermore, feeding mice an HFD decreased PGC1α mRNA levels and protein expression in skeletal muscle. However, CZH reversed these reductions. 

PGC1α deficiency also leads to abnormal oxidative fiber growth and increased body fat [42]. Muscle fibers are categorized into two types, which include slow oxidative type 1 and fast glycolytic fiber type 2 [43]. Type 1 muscle fibers are primarily used for endurance exercise, while type 2 muscles are used for short explosive movements. We determined that CZH promoted endurance exercise and increased MHC1 and MHC2A mRNA and protein levels in skeletal muscle. Therefore, increased mitochondrial biogenesis and fatty acid oxidation in skeletal muscle alleviated obesity-induced skeletal muscle atrophy and resulted in improved endurance exercise performance.

PRMTs are a group of nine enzymes that catalyze the transfer of methyl groups to target protein [44]. Dysregulation of PRMT expression or PRMT activity is associated with many prevalent health disorders, including cancer and neurodegenerative and cardiovascular diseases [45]. Recent studies have begun to describe PRMT expression and function under conditions of metabolic dysfunction. For instance, PRMT7 expression in obese mice is reduced in skeletal muscle compared to that in their lean littermates, whereas whole body PRMT7 KO animals show exacerbated age-related obesity [9]. We found that CZH increased the expression of RRMT7 mRNA and protein in the skeletal muscle of obese mice. There are no previous reports on the role of PRMT1 and PRMT6 in obesity-induced skeletal muscle atrophy. However, recent studies have reported elevated PRMT1 activity following acute exercise in mice [46] and that skeletal muscle-specific PRMT KO mice display significant reduction in muscle mass [10]. These previous studies also provided strong evidence for a crucial role of PRMT1 in muscle atrophy by identifying the PRMT1-PRMT6-FOXO3 pathway [10]. Our results showed that HFD-induced obese mice exhibited a reduction in PRMT1 levels and an increase in PRMT6 levels. Furthermore, CZH reversed the expression of RRMT1 and RRMT6 in the skeletal muscle of obese mice.

CZH is a plant rich in flavonoids, especially linarin, luteolin, and acacetin [12,47,48], which have a variety of pharmacological properties. Interestingly, our results showed that linarin and luteolin decreased mRNA levels of atrophic genes in PA-treated C2C12 cells. This was consistent with a previous report that luteolin inhibits the expression of MuRF1 and Atrogin at both the transcriptional and translational levels in skeletal muscle of a cancer-induced skeletal and cardiac muscle atrophy model [49]. Pretreatment with luteolin significantly prevents the decrease in C2C12 myotube diameter caused by LPS stimulation [50]. In a dexamethasone-induced skeletal muscle atrophy model, luteolin has a protective activity through its antioxidant and anti-apoptotic properties [51]. According to several studies, linarin makes up approximately 17% of the content of a 50% ethanol extract of CZH [52] and luteolin makes up approximately 0.5% of a methanol extract of CZH flowers [53]. Based on these results, the protective effect of CZH on PA-induced muscle atrophy may be due to synergistic effects of linarin and luteolin. Further research investigating the in vivo effects of these two main compounds of CZH is recommended.

## 4. Materials and Methods

### 4.1. Sample Preparation

CZH was purchased from Jeonnam herbal medicine farmer’s cooperative, Hwasun, South Korea. The dried CZH (0.5 kg) was soaked in 50% ethanol (5 L) at room temperature overnight. The ethanol extract was then filtered through No. 2 filter paper (Toyo Roshi Kaisha, Tokyo, Japan), concentrated under vacuum at 37 °C, and then freeze-dried. Finally, the freeze-dried CZH extracts were stored at −20 °C until use. 

### 4.2. Animals

All animal experiments were conducted in accordance with the Guidelines for Institutional Animal Care and Use Committee of the Korea Food Research Institute (KFRI-IACUC, KFRI-M-19013, approved 03/25/2019). Male C57BL/6 mice (4 w old) were maintained at a constant temperature (21–25 °C) and humidity (50%–60%) in an environment-controlled room with a 12 h light/12 h dark cycle and free access to food and water. To induce obesity, after 1 w of adaptation, the mice were fed an HFD ad libitum for 9 w (D12451; Research Diets, New Brunswick, NJ, USA) consisting of 45% calories as fat. Twenty-one HFD-induced obese mice were continued to be fed an HFD and were also orally administered either distilled water (HFD), low-dose CZH (125 mg/kg) dissolved in distilled water (125CZH), or high-dose CZH (250 mg/kg) dissolved in distilled water (250CZH) for 8 w. As a control group to compare to the HFD-fed mice, age-matched mice were fed a standard chow diet (Chow). Body weight and food intake were measured weekly. At the end of the experiments, the mice were euthanized and blood was collected. Muscle tissues were immediately removed and the weights were measured. TG levels in quadriceps muscle were also analyzed using commercial kits (Abcam, Cambridge, MA, USA) following the manufacturer’s instruction. 

### 4.3. Biochemical Analysis

The blood was collected from the abdominal aorta into a blood collection tube. Serum was subsequently obtained by centrifuging the blood at 900× *g* for 15 min at 4 °C. Commercially available kits were used to assay the serum levels of the inflammatory cytokines TNFα, MCP1 (Biolegend, San Diego, CA, USA), and IL-1β (Abcam). The serum levels of glucose (Embiel, Gyeonggi-do, South Korea), insulin (ALPCO Diagnostics, Salem, NH, USA), AST, and ALT (YD Diagnostics, Yongin, Korea) were measured by using commercial kits following the instructions of the manufacturers.

### 4.4. Treadmill and Grip Strength Tests

All mice were acclimated to running on a treadmill (Daemyoung Sci., Daejeon, Republic of Korea) for 2 d. On the first day, the mice were exposed to the treadmill with a shock grid set at a 15 ° incline and a speed of 5 m/min for 10 min followed by a speed of 10 m/min for 10 min. On the second day, the mice were subjected to running at a 15 ° incline for 5 min at a speed of 5 m/min followed by 15 min at a speed of 10 m/min. Starting the following day after the 2 d of acclimation, the mice ran for 20 min at a 15 ° incline and a speed of 10 m/min. The speed was increased by 2 m/min every 2 min. The endpoint of the speed increases was set when a mouse contacted the shock grid for 10 s. Grip strength was measured five times using a grip strength test machine (Model GS3; Bioseb, Vitrolles, France) according to the manufacturer’s instructions and the result was standardized according to body weight. The average grip strength values were calculated. The maximum and minimum values were not used. 

### 4.5. Histological and Immunohistochemical Analyses of Skeletal Muscle

Fresh gastrocnemius muscle was fixed in 4% formaldehyde, embedded in paraffin, and 4 μm sections were prepared. Gastrocnemius sections were stained with hematoxylin and eosin. Images were captured using an Olympus BX51 microscope and CSA was quantified using IMT iSolution DT 9.2 software (version 9.2, IMT i-Solution Inc., Vancouver, BC, Canada)

### 4.6. Citrate Synthase and Mitochondrial Respiratory Complex I and II Activity

To measure citrate synthase activity, homogenized quadriceps muscle tissues were centrifuged and measurements were performed using a Citrate Synthase Assay Kit (Sigma-Aldrich, St. Louis, MO, USA). The mitochondria were prepared using a Mitochondria Isolation Kit (Thermo Scientific, Rockford, IL, USA). Mitochondrial Complex I and II activity were measured in the gastrocnemius muscle tissues using a Complex I, II Enzyme Activity Microplate Assay Kit (Abcam).

### 4.7. Cell Culture

C2C12 myoblast cells (CRL-1772; ATCC, Manassas, VA, USA) were cultured in Dulbecco’s Modified Eagle Medium (DMEM) supplemented with 10% fetal bovine serum, 100 U/mL penicillin, and 100 μg/mL streptomycin at 37 °C in a humidified 5% CO_2_ atmosphere. For differentiation, the C2C12 cells were cultured in DMEM with 2% horse serum, 100 U/mL penicillin, and 100 μg/mL streptomycin for 4 d. Samples of CZH, linarin, and luteolin were treated for 24 h with 0.5 mM PA on day 4 of differentiation. PA stock solutions were prepared by conjugating PA to fatty acid-free BSA as previously described (7% BSA:5 mM PA; 5:1 molar ratio) [54]. A 7% BSA stock solution was prepared as a control.

### 4.8. Western Blot Analysis

Protein from the gastrocnemius muscle was extracted using radioimmunoprecipitation assay (RIPA) buffer. Protein concentrations of the supernatants were determined using a Pierce BCA Protein Assay Kit (Thermo Scientific, Rockford, IL, USA) and BSA as the standard. Total protein (10 μg per lane) was separated by 10% SDS−polyacrylamide gel electrophoresis and transferred to polyvinylidene difluoride membranes (Bio-Rad, Hercules, CA, USA). The membranes were blocked for 1 h at room temperature with tris-buffered saline containing 5% skim milk and 0.1% Tween 20 (Junsei, Tokyo, Japan). After overnight incubation at 4 °C with primary antibodies, the membranes were washed and incubated with a horseradish peroxidase-conjugated secondary antibody for 1 h at room temperature. Immunodetection was carried out with ECL Detection Reagent (Bio-Rad). All figures showing results of quantitative analysis using Image J software (National Institutes of Health, Bethesda, MD, USA) include data from at least three independent experiments.

### 4.9. Quantitative Real-Time PCR

Total RNA extraction from the quadriceps muscle and C2C12 cells were performed using a Qiagen RNeasy Fibrous Tissue Mini Kit (Qiagen Inc., Hilden, Germany) and NucleoSpin RNA (Macherey-Nagel, Duren, Germany), respectively. Complementary DNA (cDNA) using total RNA as the template was synthesized used a ReverTra Ace^®^ Quantitative Reverse Transcription Polymerase Chain Reaction (qPCR RT) Master Kit (Toyobo Co., Ltd., Osaka, Japan) according to the protocol provided by the manufacturer. Quantitative PCR was performed using SYBR Green Real-Time PCR Master Mix (Toyobo Co., Ltd., Osaka, Japan) and a ViiA7 PCR system (Applied Biosystems, Foster City, CA, USA). The cDNA was used as a template in 20 μL reaction mixtures that were processed using an initial step of 95 °C for 5 min, followed by 40 amplification cycles including 95 °C for 5 s; 55 °C for 10 s; and 72 °C for 15 s. The amount of each mRNA was normalized to 18 s rRNA.

### 4.10. mtDNA Content Quantitation

Total DNA was extracted using the AccuPrep Genomic DNA Extraction Kit (Bioneer, Daejeon, Korea), and qPCR was performed using mtDNA- or ncDNA-specific primers [55].

### 4.11. Statistical Analysis

Data are expressed as the mean ± SEM. Statistical analyses were performed using GraphPad Prism, version 7.04 software (GraphPad Software, San Diego, CA, USA). An unpaired *t*-test was used to assess differences between two groups (animals on standard Chow vs. HFD). One-way ANOVA was used to compare more than two groups followed by Dunnett’s multiple comparison test (HFD, 125CZH, and 250CZH).

## 5. Conclusions

We showed that CZH ameliorated obesity-induced skeletal muscle atrophy in HFD-fed mice. This beneficial effect involved the regulation of PRMT1, PRMT6, and PRMT7, and the attenuation of the mitochondrial dysfunction. These results suggest that CZH may have a beneficial effect on the prevention and treatment of skeletal muscle atrophy in obesity. However, further clinical studies need to be carried out to confirm our findings in humans, and the results of this research will provide a basis for these clinical trials. In addition, further studies using healthy mice and aged mice are also recommended to have a better understanding of the effect of CZH on skeletal muscle atrophy.

## Figures and Tables

**Figure 1 ijms-21-02811-f001:**
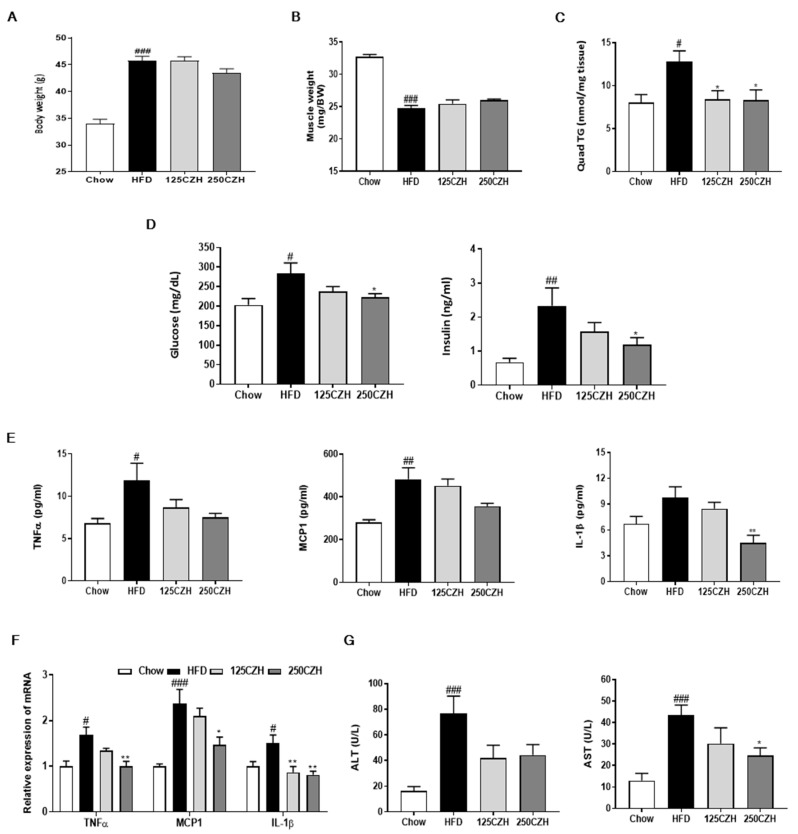
Effect of *Chrysanthemi zawadskii* var. *latilobum* (CZH) on obesity-induced skeletal muscle fat accumulation and metabolic parameters in C57BL/6 mice. Male C57BL/6 mice fed a high-fat diet (HFD) for 9 w to induce obesity. The diet-induced obese mice were then maintained on an HFD and orally administered either distilled water (HFD), low-dose CZH (125 mg/kg; 125CZH), or high-dose CZH (250 mg/kg; 250CZH) for 8 w. Age-matched mice were fed a standard chow diet (Chow) as a control group for comparison. (**A**) Body weights of the experimental mice. (**B**) Skeletal muscle weights of the experimental mice. (**C**) Triglyceride (TG) levels in quadriceps muscle of the experimental mice. The serum levels of (**D**) glucose, insulin, and (**E**) inflammatory cytokines, including TNFα, MCP1, and IL-1β in the experimental mice. (**F**) *TNFα*, *MCP1*, and *IL-1β* mRNA levels in the quadriceps muscle of the experimental mice were analyzed by qRT-PCR. **(G)** The serum alanine aminotransferase (ALT) and aspartate aminotransferase (AST) levels of the experimental mice. Results are expressed as mean ± SEM (*n* = 8). * *p* < 0.05, ** *p* < 0.01 versus the HFD group. # *p* < 0.05, ## *p* < 0.01, ### *p* < 0.001 versus the Chow-fed group.

**Figure 2 ijms-21-02811-f002:**
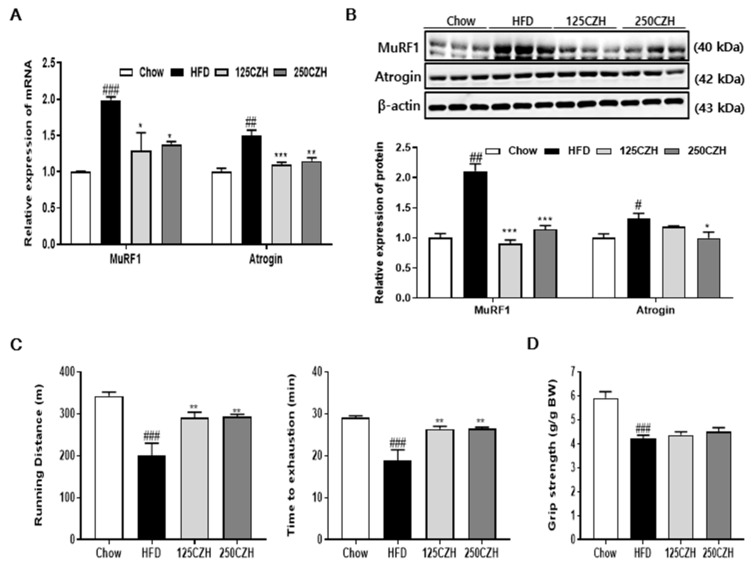
Effect of CZH on obesity-induced skeletal muscle atrophy in HFD-fed mice. (**A**) Expression levels of muscle atrophy-related genes muscle RING-finger protein 1 (MuRF1) and Atrogin in quadriceps muscle quantified by qRT-PCR. (**B**) Expression levels of MuRF1 and Atrogin in gastrocnemius muscle quantified by Western blotting. Band density was analyzed using ImageJ software. (**C** and **D**) Exercise capacity was measured according to running distance (m), time (min), and grip strength (g/g BW). Results are expressed as mean ± SEM (*n* = 8). * *p* < 0.05, ** *p* < 0.01, *** *p* < 0.001 versus the HFD group. # *p* < 0.05, ## *p* < 0.01, ### *p* < 0.001 vs. the Chow-fed group.

**Figure 3 ijms-21-02811-f003:**
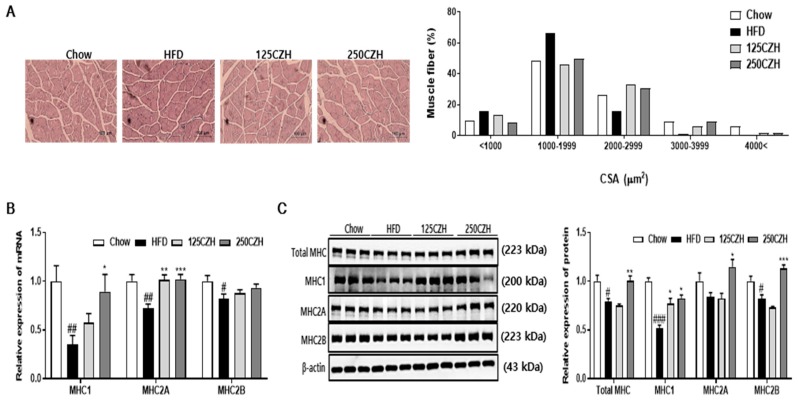
Effect of CZH on muscle fiber size and type in mice. (**A**) Representative image of myofiber cross-section of gastrocnemius muscle (hematoxylin and eosin staining; scale bar, 100 μm). Images were analyzed by light microscopy, and cross-sectional area (CSA) was measured. Distribution of muscle fiber CSA (*n* = 3 per group). (**B**) qRT-PCR analyses of MHC1, MHC2A, and MHC2B mRNA expression in quadriceps muscle. (**C**) Expression of total MHC, MHC1, MHC2A, and MHC2B in gastrocnemius muscle as determined by Western blotting. Band density was analyzed using ImageJ software (National Institutes of Health). Results are expressed as mean ± SEM (*n* = 8). * *p* < 0.05, ** *p* < 0.01, *** *p* < 0.001 versus the HFD group. # *p* < 0.05, ## *p* < 0.01, ### *p* < 0.001 vs. the Chow-fed group.

**Figure 4 ijms-21-02811-f004:**
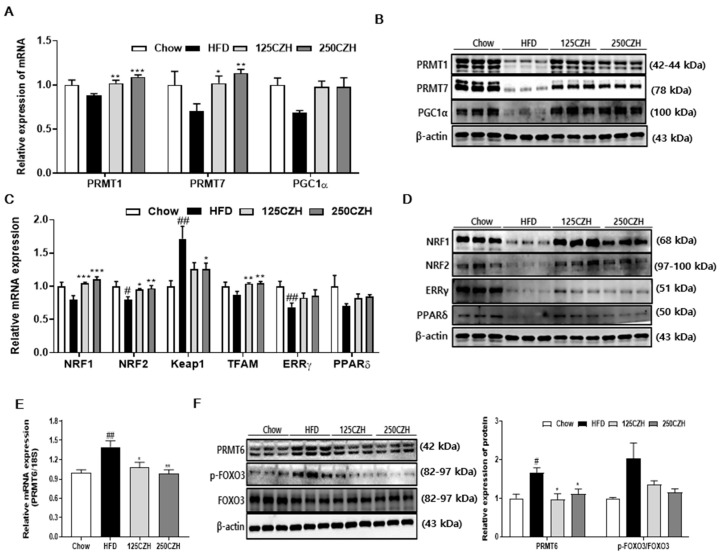
Effect of CZH on regulation of skeletal muscle PRMTs in mice. (**A**) Expression levels of PRMT1, PRMT7, and PGC1α genes in quadriceps muscle quantified by qRT-PCR. (**B**) Expression levels of PRMT1, PRMT7, and PGC1α genes in gastrocnemius muscle quantified by Western blotting. (**C**) qRT-PCR analyses of NRF1, NRF2, Keap1, TFAM, ERRγ, and PPARδ mRNA expression in quadriceps muscle. (**D**) Expression of NRF1, NRF2, ERRγ, and PPARδ in gastrocnemius muscle as determined by Western blotting. (**E**) PRMT6 mRNA levels in quadriceps muscle. (**F**) Expression of total PRMT6, p-FOXO3, and FOXO3 in gastrocnemius muscle as determined by Western blotting. Band density was analyzed using ImageJ software. Results are expressed as mean ± SEM (*n* = 8). * *p* < 0.05, ** *p* < 0.01, *** *p* < 0.001 versus the HFD group. # *p* < 0.05, ## *p* < 0.01, ### *p* < 0.001 versus the Chow-fed group.

**Figure 5 ijms-21-02811-f005:**
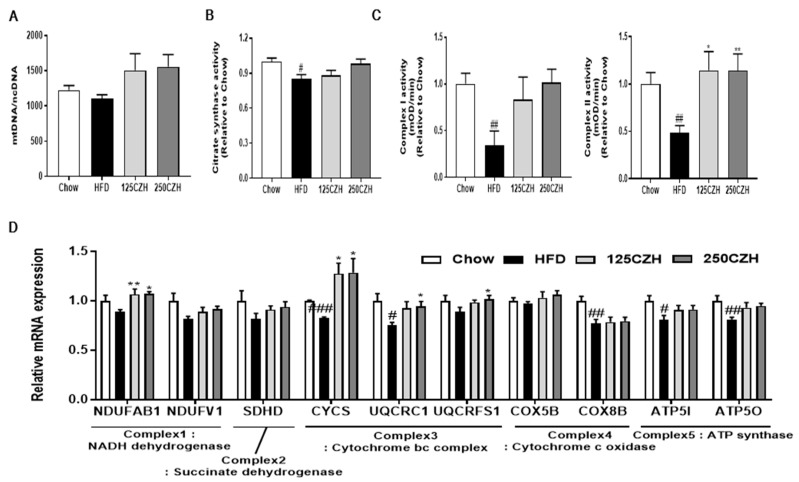
Effect of CZH on mitochondrial dysfunction in obese mice. (**A**) Mitochondrial DNA (mtDNA) content, **(B)** citrate synthase activity in quadriceps muscle of the mice. (**C**) Complex I and Complex II enzyme activities in gastrocnemius muscle of the mice. (**D**) Relative transcript levels in quadriceps muscle of genes encoding proteins involved in OXPHOS. Results are expressed as mean ± SEM (*n* = 8). * *p* < 0.05, ** *p* < 0.01, *** *p* < 0.001 versus the HFD group. # *p* < 0.05, ## *p* < 0.01, ### *p* < 0.001 vs. the Chow-fed group.

**Figure 6 ijms-21-02811-f006:**
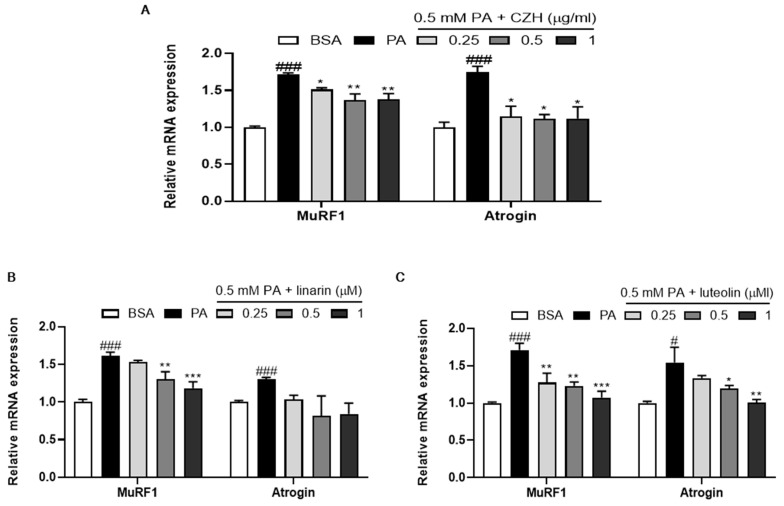
Effect of CZH, linarin, and luteolin on palmitic acid (PA)-induced muscle atrophy in C2C12 cells. The cells were treated with 0.5 mM PA for 24 h in the presence or absence of CZH, linarin, and luteolin on day 4 of differentiation. (**A**–**C**) Expression levels of muscle atrophy-related genes quantified by qRT-PCR for MuRF1 and Atrogin in PA-treated myotubes. Results are expressed as mean ± SEM from three independent measurements. * *p* < 0.05, ** *p* < 0.01, *** *p* < 0.001 versus PA-treated myotubes. # *p* < 0.05, ## *p* < 0.01, ### *p* < 0.001 vs. the BSA-treated myotubes.

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
