# Peer review of "Chrysanthemi Zawadskii var. Latilobum Attenuates Obesity-Induced Skeletal Muscle Atrophy via Regulation of PRMTs in Skeletal Muscle of Mice"

_ijms, 2020, doi:10.3390/ijms21082811_

Round 1

Reviewer 1 Report

In this study by Yoo A. et al. the effects of Chrysanthemi zawadskii var. latilobum extract on obesity induced skeletal muscle atrophy were investigated. CZH ethanol extract was able to recover the obesity related phenotype as well as muscle atrophy by improving function. CZH 250 mg/kg bd dose can improve inflammation as well as mitochondrial function. The manuscript is well written, presented nicely and well detailed. The students should be appreciated. However, there are few concerns regarding this study and manuscript.

Major-specific comments:
1. As soon as I have started reading the manuscript, I was looking for the statement that why this study has been conducted at the first place? There is already a dozen of study published that, how mitochondria/metabolism/function can be improved in obesity (specifically – NAD precursors, Resveratrol, metformin etc.). There are studies also looking at maternal obesity with NMN treatment and improvement in exercise capacity. So, why we need a new natural extract (not compound!!) to treat that obesity related issues? The authors have mentioned in the introduction “However, the effect of CZH on obesity induced skeletal muscle atrophy has not been reported” I do not believe this statement can be a scientific reason to investigate or conduct a research.
2. Obesity is major non-communicable diseases, which is causing many additional metabolic abnormalities including, mitochondrial function, biogenesis and inflammation. The authors discussing mainly inflammation, muscle function however not touch the mitochondrial function, although manuscript shows many mitochondrial functional parameters. I would like to see a bit more discussion about the mitochondrial function rather than muscle function. Provided, the authors claiming muscle atrophy is one of the muscle dysfunctions, but it is mitochondrial dysfunction.
3. CZH ethanol extract was used in this study. Why ethanol extract? What part of the herb? Why not isolated compound?
4. How the authors came up with 125 or 250 mg/kg dose? What about 500mg/kg BW?
5. In obesity or exercise related studies (such as mentioned above NMN study with obesity and exercise comparison), first thing to look at is the glucose tolerance test or insulin resistance. Did the authors look at those parameters? I can understand if the data were not taken it is hard to reproduce the colony but at least (if GTT or ITT not available) what about the plasma glucose level or insulin level in those animals?
6. The authors discussed about the mtDNA, however, the authors did not measure it, it can be measured easily using rtPCR. Is there any reason for not measuring mtDNA when the authors making a huge point regarding this?
7. Did the authors think about metabolic markers (as the authors did not measure GTT or glucose, I would imagine they did not think about it), such as mitochondrial metabolism targets improved, provided, mitochondrial function improved as shown in figure 5.
8. I think the authors should reconsider the term muscle function or mitochondrial function.
9. Why the authors did not investigate CZH treatment effect in chow fed animal?
10. Discussion does not provide any therapeutic emphasis. This discussion mostly re-explained the result. How this research outcome can be incorporated in the future research or treatment?
11. The conclusion should be re written – “These results indicate that CZH may have a beneficial effect on the prevention and treatment of skeletal muscle atrophy.” It should be - These results indicate that CZH may have a beneficial effect on the prevention and treatment of skeletal muscle atrophy in obesity. As the authors did not show any benefits in chow group.
12. Statistics – all figures showing significant sign in CHOW bar, this does not make sense. As the significance is to measure if the HFD group is higher or lower (as authors showing the CZH treated groups compared to HFD).

Minor-specific comments:
1. Please make sure the absolute value of inflammatory markers is correct figure 1D. It seems, TNFa is 10-fold lower than other published levels.
2. Commercially available kits were used for cytokines, however, how the samples were processed (eg. Centrifugation speed)? From my experience, if collected blood centrifuged more than 3000g for 10 min then you wouldn’t see real value for the cytokines. May consider make a separate section 4.3 for the method.
3. Prism, one way is not a statistical analysis. May be the authors meant one-way ANOVA
4. Reconsider making figure legends little bigger. Figure 5C all those gene I can read clearly, however, no other figure has larger legends. Hard to read.
5. First line of the introduction needs a reference.
6. Line number 40, “which results in impaired skeletal muscle”, did you mean function?
7. Figure 1D size is different than other figures, please try to make same size or readable.
8. Western blots representative blots, some of them are very over exposed, is there any lighter exposure? I understand there are some figures you need to show exposed to show other groups but please see following -
- Figure 2B MuRF1
- Figure 3C MHC1
- Figure 4D – NRF1, NRF2 and ERR
- Figure 4F FoXO3
9. What was the research diet number? Line#273

Author Response

Dear Reviewer

We would like to thank you for insightful and constructive comments. These comments helped us strengthen and clarify our manuscript. We have revised our manuscript in response to your commnents. Please find attache file.

Reviewer 2 Report

In this manuscript, Yoo et al. reported the beneficial effects of Chrysanthemi zawadskii var. latilobum (CZH) in the attenuation of obesity-induced skeletal muscle atrophy via regulation of PRMTs in skeletal muscle of mice. The authors performed several experiments to prove their proposed hypothesis. The generated data were analyzed appropriately and also discussed reasonably. Overall this is a good piece of study in this field. However, the authors need to address the following concerns while revising the manuscript.

Comments:

  1. Provide relative molecular weight for each protein studied in this study.
  2. Did the authors study the expression of Keap1, a negative regulator of Nrf2 as well as the downstream proteins of Nrf2 signaling?
  3. Provide nuclear expression of Nrf2 in this proposed study.
  4. Did CZH has any toxicity? Explain.

Author Response

Dear Reviewer

We would like to thank you for insightful and constructive comments. These comments helped us strengthen and clarify our report. We have revised our manuscript a point by point in response to the your comments as noted below

In this manuscript, Yoo et al. reported the beneficial effects of Chrysanthemi zawadskii var. latilobum (CZH) in the attenuation of obesity-induced skeletal muscle atrophy via regulation of PRMTs in skeletal muscle of mice. The authors performed several experiments to prove their proposed hypothesis. The generated data were analyzed appropriately and also discussed reasonably. Overall this is a good piece of study in this field. However, the authors need to address the following concerns while revising the manuscript.

Comments:

  1. Provide relative molecular weight for each protein studied in this study.

Response 1. We added relative molecular weight for all proteins in the revised all figures

  1. Did the authors study the expression of Keap1, a negative regulator of Nrf2 as well as the downstream proteins of Nrf2 signaling?

Response 2. Thank you for your comment. We analyzed Keap 1 expression as you pointed out, and added to Figure 4 and result 2.4 (Line# 155)

  1. Provide nuclear expression of Nrf2 in this proposed study.

Response 3.Unfortunately, we didn’t analyze nuclear expression of Nrf2. However, we are planning to analyze this in our next study

  1. Did CZH has any toxicity? Explain.

Response 4. Any toxicity of CZH has not been reported. And, we did not observed any abnormality in organs when mice were sacrificed. In addition, we analyzed ALT and AST in serum and those enzyme levels were decreased in CZH group compared with HFD group.    

We added serum ALT and AST to Figure 1G and text (Line# 83)

Round 2

Reviewer 1 Report

Replies are appropriate and satisfactory. 

Reviewer 2 Report

The authors revised the manuscript satisfactorily.